# Juliet: Per-Sample Conditional Branching for Efficient Convolutional Networks

## Abstract

We introduce Juliet, a dynamic, trie-augmented neural architecture that improves the efficiency of convolutional neural networks by routing each input through learned per-node branches while growing and pruning capacity on the fly. Each node pairs a lightweight sub-module with a transformer-based path selector trained end-to-end; growing and pruning based on exponential moving average (EMA) usage let the model expand or contract during training to preserve accuracy within compute and memory budgets. We graft Juliet onto ResNet-18, EfficientNet-B0, and DenseNet-121 and train on CIFAR-10 (ARCHER2), with an ImageNet/H100 check using ResNet-101. On CIFAR-10, Juliet reduces theoretical training and inference FLOPs, even when the parameter count increases. The results show a $\sim 21\%$, (ResNet-18), $\sim 68\%$ (EfficientNet-B0), and $\sim 70\%$ (DenseNet-121) in inference flops, while staying within $\sim 1\%$ Top-1 of the baseline for ResNet-18 and DenseNet-121, with a larger trade-off on EfficientNet-B0. At ImageNet scale, Juliet-101 achieves 27.1 Top-1 per GFLOPs, outscoring SkipNet, ConvNet-AIG, and BlockDrop. Ablations and hyperparameter sweeps (growth/prune thresholds, prune interval, prebuild limit) reveal nuances in Juliet's architecture, and simpler routers (e.g., a small MLP) match transformer routing, indicating the transformer router may not be a prerequisite for achieving competitive accuracy. Overall, Juliet provides a flexible, interpretable approach to conditional computation for convolutional neural networks, improving the efficiency–accuracy trade-off for the CNNs we evaluate.

## 1 Introduction

Neural networks come in different sizes, from massive models that require robust datacenter infrastructure to run, to compact versions optimized to run efficiently on smartphones and other mobile devices, yet most production models remain *static*: once trained, they process every input through the same full stack of layers. This uniform path ignores input difficulty and hardware variability, wasting computation and limiting efficiency.

Dynamic neural networks address this by adapting computation to the sample, space, or time dimension. Sample-wise models tailor the path per input; spatial models gate computation by location (common in vision); temporal models vary effort across time for speech and video. See (32) for a survey and taxonomy of these directions. Despite progress, many approaches still rely on hand-tuned routing rules, pre-defined mechanisms, or focus mainly on inference-time savings rather than architectures that *learn* their own structure during training.

This work introduces *Juliet*, a sample-wise dynamic architecture designed to couple learned routing with online topology evolution:

1. **Trie-structured layers.** Layers are organized as nodes in a trie to enable prefix sharing and flexible branching.

2. **Per-node learned routing.** Each node hosts a lightweight transformer-based selector that routes using intermediate features.

3. **Grow–prune dynamics.** Capacity evolves online via variance-driven split triggers (growth) and EMA-based usage pruning (shrinkage). Hyperparameters such as growth threshold, prune threshold, and prune interval shape the topology over training.

We study Juliet's computational efficiency and accuracy on convolutional neural networks under controlled single-device settings on the University of Edinburgh's systems: one CPU on ARCHER2 and one NVIDIA H100 (80 GB HBM3) on EIDF. Experiments probe training and inference efficiency, architecture versatility across standard CNN backbones, and sensitivity to growth/prune hyperparameters.

Our goals are twofold: (i) determine whether Juliet can materially reduce computation without degrading accuracy, and (ii) test whether transformer-based routing enables networks to learn effective computational topologies end-to-end.

**Contributions.**   The contributions of this paper are:

- A trie-structured, sample-wise dynamic architecture with per-node learned routing and online grow–prune mechanics.

- A practical training recipe that stabilizes topology learning under standard supervised losses.

- An empirical study across ResNet-18, EfficientNet-B0, and DenseNet-121 on CIFAR-10 (ARCHER2/EIDF), plus an ImageNet/H100 sanity check with ResNet-101, including FLOP, profiling and hyperparameter sensitivity.

- An analysis of utilization and routing overheads explaining gaps between theoretical and realized speedups, with deployment guidelines.

## 2   Background

Neural networks underpin modern AI from language models to vision yet training and inference remain resource-intensive. Empirical scaling laws show that increasing parameters, data, and compute reduces loss, pushing budgets upward as models grow (42). Meanwhile, most production models are *static*: every input traverses the same layers, regardless of difficulty, which wastes computation and limits deployment flexibility. Dynamic neural networks address this by *conditioning* computation on the input—skipping layers, altering depth/width, or routing through branches—rather than processing all inputs identically (32; 50). This perspective echoes the adaptive processing of biological systems (54).

### 2.1   Core Motivations for Dynamic Neural Networks

Dynamic models allocate less compute to easy samples and more to hard ones via early exits or layer skipping (69; 73). If a static network costs $C_s$ per input, and a dynamic model spends $C_e < C_s$ on an "easy" fraction $p_e$ and $C_h \geq C_s$ on the remainder, the expected cost is

$$\mathbb{E}[C] = p_e C_e + (1 - p_e) C_h < C_s \quad \Rightarrow \quad \text{speedup} = \frac{C_s}{\mathbb{E}[C]} > 1,$$

with empirical gains reported under input-dependent execution (26).

Dynamic networks trade accuracy for latency under resource constraints and scale up on servers, adapting online to data and hardware conditions (48; 9; 72). This enables a single model family to target phones, edge devices, and clusters by tuning routing/halting policies.

Conditional computation decouples parameter count from per-sample cost. Mixture-of-Experts (MoE) activates only a subset of experts per input, supporting very large models with manageable compute (64; 47; 21). Input-conditioned execution also enlarges the function class representable by a fixed-parameter template (8).

Instance-aware routing preserves performance on hard inputs while saving FLOPs on easy ones; routing can encourage specialization and richer representations (60; 37).

Many dynamic techniques are model and domain agnostic and mirror selective attention in natural systems, potentially aiding interpretability by exposing which subpaths fire for which inputs (32; 41; 49; 77).

## 2.2 Key Limitations of Dynamic Neural Networks

Discrete or sharply nonlinear routing (e.g., token-to-expert assignment, hard layer skips) can induce gradient spikes and divergence; stability often hinges on careful objectives, regularizers, and curricula (21; 75; 20). A straight-through surrogate for a binary gate can flip decisions with small logit changes, derailing optimization.

Underpowered gates may choose suboptimal experts or collapse to few paths, reducing accuracy and parallelism; routing also adds compute and communication that can erode theoretical gains (51; 8; 72; 47).

Kernel launch costs, irregular control flow, and path variance reduce measured speedups relative to FLOP savings (32; 47; 20). For a batch of size $B$ with per-sample path cost $C_i$,

$$T_{\text{batch}} \approx \max_{i \leq B} C_i \; + \; \text{overhead},$$

so high variance in $C_i$ hurts device occupancy even when total FLOPs drop.

GPUs/TPUs are optimized for large, dense, and regular workloads. Input-dependent sparsity yields partial occupancy and irregular memory access, depressing throughput (56; 32). Consequently, the measured speedup often trails the theoretical reduction in FLOPs.

Gates can overuse a small subset of experts, harming learning and parallel speed. Load-balancing losses are commonly added to discourage collapse (21; 60). For experts $m = 1 \ldots M$ receiving fractions $f_m$, a simple penalty

$$\mathcal{L}_{\text{load}} = M \sum_{m=1}^{M} \left( f_m - \tfrac{1}{M} \right)^2$$

encourages equitable routing during training.

Conditional paths can hide bottlenecks: different inputs (or runs) traverse different subgraphs, complicating profiling, regression analysis, and reliability efforts (32; 50).

Dynamic models jointly learn weights and routing, creating chicken-and-egg dynamics: a gate avoids an unproven expert, and the expert never improves without traffic. The resulting bilevel landscape may slow convergence or yield suboptimal fixed points; nondeterministic paths further complicate replication (8; 73; 20; 32).

Static, one-size-fits-all computation fails to reflect input variability and deployment constraints. Dynamic neural networks address this by conditioning computation on the sample, improving efficiency, flexibility, and capacity while maintaining accuracy. Realizing these benefits at scale, however, demands robust routing objectives, careful systems design to minimize overhead, hardware-aware implementations to close utilization gaps, and practices that improve stability and reproducibility (32; 20).

## 3 Existing Work

Dynamic neural networks have been explored through various mechanisms that adapt model computation to the input or task. We briefly review key approaches, including adaptive computation, dynamic routing, architecture morphing, input-dependent execution paths, and recurrent dynamic structures.

**Adaptive Computation.** One direction is to adjust a network's depth or computation on a per-input basis. *Early-exit* architectures insert auxiliary classifiers at intermediate layers so that "easy" inputs can exit with a prediction before reaching the end of the network. BranchyNet (69) is a seminal example that adds side branches to a deep network, allowing confident predictions to exit early and thereby saving computation.

Similarly, Shallow-Deep Networks (43) address the issue of *network overthinking* by letting easy instances be handled by shallow portions of the model while only difficult cases traverse the full depth. Another approach, *Adaptive Computation Time* (ACT), introduces a dynamic halting mechanism for recurrent networks. In ACT, each input at a recurrent layer can adaptively decide whether to continue processing or halt. Graves (26) first proposed ACT for RNNs, enabling the network to learn how many computational steps to allocate for each input sequence, effectively varying its "depth" (in time) based on the input's complexity.

**Dynamic Routing.** Another class of dynamic networks use learned routing to activate only parts of a model for each input. *Mixture-of-Experts* (MoE) models are a prime example: a gating network assigns each input to one or a few among many expert subnetworks, so that only those "experts" are executed. This idea dates back to early work by Jacobs *et al.* (39), who introduced adaptive mixtures of local experts. Modern deep MoEs, such as the sparsely-gated MoE of Shazeer *et al.* (64), have demonstrated the ability to scale up model capacity while keeping per-input computation constant by activating only a small fraction of model parameters for each sample. Beyond MoEs, *Capsule Networks* (63) perform dynamic routing at the level of neuron groups ("capsules"). In a capsule network, low-level capsules vote for higher-level capsules via a routing-by-agreement mechanism: an iterative routing procedure determines how outputs of lower-level capsules are passed to higher-level ones based on agreement between their predictions. This dynamic routing of information between layers allows capsule networks to learn part-whole relationships and achieve viewpoint invariance, a behavior that static convolutional networks struggle to capture.

**Architecture Morphing.** Dynamic networks can also adapt their *architecture* over time or across tasks. *Neural Architecture Search* (NAS) techniques automate the discovery of network structures, sometimes yielding dynamic architectures. Zoph and Le's seminal work (79) used reinforcement learning to search for convolutional architectures, demonstrating that learned architectures can outperform manually designed ones. Subsequent methods like DARTS (51) made NAS more efficient by relaxing the search to a differentiable optimization problem, effectively learning architecture parameters via gradient descent. While NAS generally operates offline, it lays the groundwork for models that can morph structure. In parallel, researchers have explored algorithms to grow or prune network components during training or deployment. *Progressive Neural Networks* (62) exemplify dynamic growing of architecture in a continual learning setting: as new tasks arrive, the network expands by adding new "columns" (layers for the new task) while preserving and leveraging the previously learned features via lateral connections. This allows accumulation of knowledge without catastrophic forgetting, at the cost of a larger network over time. On the flip side, pruning approaches aim to dynamically remove unnecessary parts of the network to improve efficiency. For instance, Han *et al.* (2015) introduced a pruning method for learned networks that removes weights below a certain threshold, yielding a sparser model (30). Contemporary research extends this idea to runtime pruning, where the model can selectively drop neurons or filters for particular inputs or conditions, effectively changing its architecture on the fly to save computation.

**Input-Dependent Execution Paths.** A related line of work focuses on skipping computations based on each input's characteristics, often in the context of deep CNNs. SkipNet (73) learns gating networks to decide, at runtime, whether to execute or bypass certain layers in a ResNet-like architecture. By routing challenging inputs through more layers and allowing easier inputs to skip blocks, SkipNet achieves significant inference speedups while maintaining accuracy. In a similar vein, BlockDrop (74) uses a reinforcement learning agent (a policy network) to dynamically choose which residual blocks to execute for a given input image. BlockDrop was shown to reduce computation on image classification tasks by learning to omit blocks that are less important for the current sample, effectively finding an input-dependent path through the network. Beyond skipping whole layers, other methods make the layer operations themselves dynamic. *Dynamic Filter Networks* (16) generate filters as a function of the input, so that the convolutional kernels used at runtime are adaptively crafted for each sample. Instead of having a fixed set of learned filters, a dynamic filter network has a small sub-network (often conditioned on some aspect of the input) produce the convolution weights, which are then applied to the input. This yields a form of content-based dynamic execution, where the model can emphasize different features or patterns depending on the input, effectively changing its computations on the fly. Techniques like SkipNet, BlockDrop, and dynamic filters all illustrate the principle of *conditional computation*: they strive to spend less computation on easy or irrelevant parts of the input, and more on the important or challenging parts, thereby improving efficiency without sacrificing performance on hard cases.

**Recurrent Dynamic Structures.** The concept of dynamic networks extends to models that incorporate memory and can alter their computation across time. *Neural Turing Machines* (NTMs) (25) were an early attempt at coupling neural networks with an external memory tape, allowing the model to learn algorithms like copying or sorting by reading from and writing to memory locations in a data-dependent manner. The NTM's controller is a neural network that decides, at each time step, how to interact with the memory (via differentiable read/write heads), effectively learning a program. This idea was later significantly advanced in the *Differentiable Neural Computer* (DNC) by Graves *et al.* (25), which augmented the memory architecture and demonstrated the ability to solve complex tasks like graph traversal and question answering by dynamically storing and retrieving information. These memory-augmented networks embody dynamic behavior in that they can flexibly allocate computational resources (memory slots, read/write operations) based on the input sequence and intermediate results. The "route" an input takes is not through different feed-forward layers, but through different memory access patterns and internal states. Such models blur the line between hard-coded algorithmic logic and learned neural computation, pointing toward networks that can not only adjust how much computation to use, but also learn discrete-like operations (like reading/writing to specific memory addresses) conditioned on the data. This opens the door to neural models that inherently adapt their inference procedure – a dynamic control flow – as exemplified by NTMs, DNCs, and related architectures in the realm of neural programming.

### 3.1 Juliet Architecture: Specifications

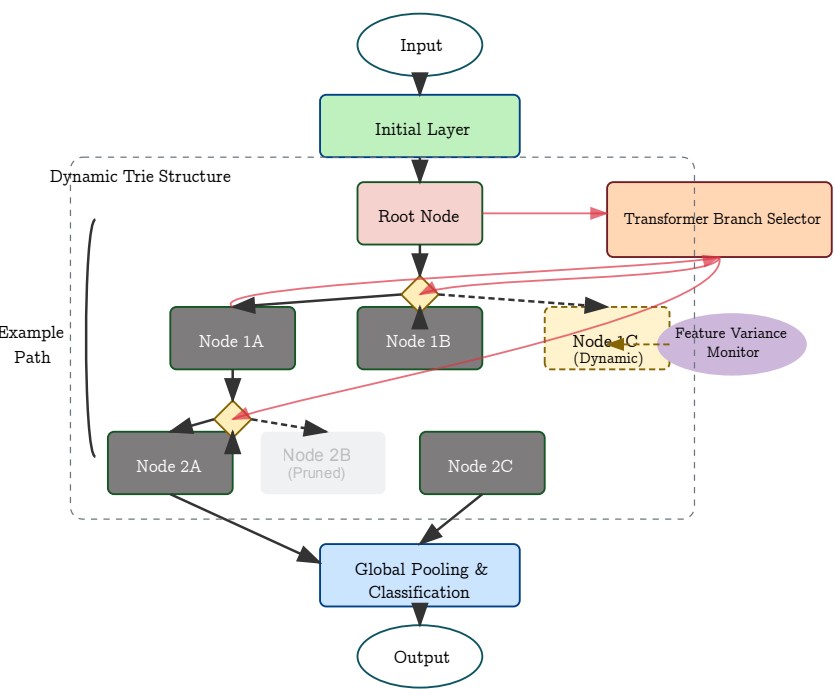

Figure 1: Juliet: trie-structured dynamic network with per-node routing.

Juliet is a *trie-augmented* neural network: instead of a fixed, monolithic stack, layers are organised as nodes in a trie that exposes multi-way branching and prefix sharing (10; 1) (See Figure 1). Each node hosts (i) a small neural submodule (e.g., residual block/MLP) and (ii) a learned *branch selector*. Inputs traverse a single root-to-leaf path chosen conditionally on intermediate features; branches can be *grown* or *pruned* online, yielding a compact, specialised topology over training.

Given a node representation $h^{(n)} \in \mathbb{R}^d$, the selector emits branch probabilities

$$p^{(n)} = \mathrm{softmax}\big(W^{(n)}h^{(n)}\big), \quad p_k^{(n)} = \Pr(\text{branch } k \mid h^{(n)}), \tag{1}$$

and forwards to one of the $K_n$ children using top 1 deterministic routing. Unlike simple binary gates, Juliet's transformer-based selector (§3.1.2) can attend to richer context, improving decision quality (71).

**Why a trie (vs. generic trees/graphs)?** Trie structure naturally supports multi-way branching with shared prefixes and selective activation, and it aligns with online *grow–prune* operations that change branching factors locally without refactoring the whole graph. Prior trie-augmented networks embed learning into the hierarchy (1); Juliet adopts this idea at the *block* granularity (not full networks per node), enabling scalable training and clearer path semantics.

### 3.1.1 Dynamic Growth and Pruning

Juliet adapts capacity during training.

*Split (grow) trigger.* For node $n$, track activation variance

$$\sigma_n^2 = \frac{1}{B}\sum_{i=1}^{B}\big\|h_i^{(n)} - \bar{h}^{(n)}\big\|_2^2, \quad \bar{h}^{(n)} = \tfrac{1}{B}\sum_i h_i^{(n)}. \tag{2}$$

*Prune trigger.* Maintain an EMA of branch usage and remove long-idle branches (4):

$$a_t^{(n)} = \beta a_{t-1}^{(n)} + (1 - \beta)\,\mathbf{1}\{\text{branch } n \text{ taken}\}, \quad \text{prune if } a_t^{(n)} < \theta_{\text{prune}} \text{ for } t \geq t_{\min}. \tag{3}$$

This minimalist-to-specialised schedule stabilises training compared to fixing a large dynamic topology a priori (50; 1).

### 3.1.2 Transformer-Based Routing and Conditional Execution

At each node, Juliet employs a lightweight transformer router. Attention offers expressive routing, akin to conditional computation in Transformer MoEs (21), but here, decisions are *local and hierarchical* rather than global and flat. Router cost scales with hidden size and head count but remains modest relative to the active path's compute, since only one path's routers are evaluated per input.

### 3.1.3 Addressing Dynamic-NN Challenges

Juliet's design choices target known pain points (instability, routing quality, overhead, utilisation, imbalance, convergence, reproducibility).

Transformers yield higher-fidelity decisions than linear gates (71), trading mild per-node cost for better path selection. Hierarchical decomposition avoids one massive global router.

Single active path reduces dynamic kernels per sample; annealing Gumbel temperature moves routing towards near-deterministic at inference; static compilation/masking and batching per depth can further amortise costs (50; 40; 47).

Branch pruning concentrates traffic on useful paths; depth-wise grouping admits pipeline-style execution across devices in principle (38). Custom kernels (as in MoE systems) can improve gather/scatter where needed (47).

## 4 Algorithm

We describe the unified algorithm for the Juliet architecture. Each node holds a lightweight sub-module and a learnable *path selector.* During a forward pass, the input is routed from the root to a single leaf; the routed

Table 1: Juliet in context (capabilities).

| Method family | Dynamic depth | Per-node router | Grow & prune | Trie structure |
|---|:---:|:---:|:---:|:---:|
| Flat MoE (Switch/Hash)[a] | No | No | No | No |
| Early-exit / Skip (AIG, BlockDrop)[b] | Yes | No | No | No |
| Hierarchical MoE / Trees (non-trie)[c] | Yes | Partial | Partial | No |
| **Juliet (this work)** | **Yes** | **Yes** | **Yes** | **Yes** |

[a] Global router over flat experts; fixed depth.  [b] Conditional exits/skips; no topology learning.  [c] Hierarchies without online grow–prune or trie organisation.

path determines the executed computation. The structure can *grow* (by adding children) when uncertainty is high, and *prune* (by disabling or removing children) when usage is low.

Given a backbone organised as a sequence of blocks $\{b_\ell\}_{\ell=1}^L$, we *wrap* selected blocks with trie nodes. A node $N$ owns a sub-module $N$.sub (same family/type as the wrapped block) and a selector $N$.PS. Children are stored in $N$.children.

The following notation is used throughout the section:

| | |
|---|---|
| $T$ | the trie (rooted at $N_{\text{root}}$) |
| $N$ | a node with depth $N$.depth $\in \{0, \ldots, d_{\max}\}$ and $k_N = |N.\text{children}|$ |
| $N$.sub | sub-module executed at $N$ |
| $N$.PS | Transformer path-selector attached to $N$ |
| $\alpha_C$ | EMA usage score for child $C$ (per parent) |
| $\sigma_N^2$ | running activation-variance estimate at $N$ |
| $\theta_{\text{grow}}, \theta_{\text{prune}}$ | grow / prune thresholds |
| $E_{\text{prune}}$ | pruning-sweep interval (epochs) |

### 4.1 Forward Pass and Routing

**Selector.** Each selector is a small `TransformerEncoderLayer` with model width $d_{\text{model}}$ and $n_{\text{head}}$ heads. We form: (i) a *feature token* by global-average pooling the current activations and projecting to $d_{\text{model}}$ after LayerNorm; and (ii) one *branch token* per candidate child by extracting a compact weight signature from the child's sub-module (e.g., averaging a representative convolution's weights) and projecting to $d_{\text{model}}$. Concatenating [feature ∥ branches] and applying the Transformer yields branch representations; a linear head produces logits and a softmax gives $P \in \mathbb{R}^{k_N}$.

**Routing policy.** Both training and inference use **deterministic Top-1** routing: $i^\star = \arg\max_i P_i$. After each decision we update a per-child EMA usage score.

---

**Algorithm 1** Recursive forward pass with Top-1 routing and EMA update

---

**Require:** input $x$, node $N$
**Ensure:** features $f$ after traversing the subtree
 1: **function** FORWARDPASS($x, N$)
 2:     $f \leftarrow N.\text{sub}(x)$
 3:     **if** $N$.children $= \emptyset$ **then return** $f$
 4:     **else**
 5:         $P \leftarrow \text{softmax}(N.\text{PS}(f)) \in \mathbb{R}^{k_N}$
 6:         $i^\star \leftarrow \arg\max_i P_i$                     ▷ Top-1 (deterministic)
 7:         **for** $j \in \{1..k_N\}$: $\alpha_{N.\text{children}[j]} \leftarrow (1-\lambda)\,\alpha_{N.\text{children}[j]} + \lambda\,\mathbf{1}[j = i^\star]$
 8:         **return** FORWARDPASS($f$, $N.\text{children}[i^\star]$)
 9:     **end if**
10: **end function**

---

**Channel adapter (optional).** Because different leaves may output different channel sizes, we optionally insert a $1{\times}1$ *adapter* that maps the current channels to a fixed head width before global average pooling and the classifier. Adapters are cached per observed channel count.

## 4.2 Dynamic Growth

Growth is attempted periodically during training (e.g., every $g$ mini-batches). We first traverse from the root using the same Top-1 policy to the deepest active node on the current input, then evaluate a mini-batch uncertainty proxy.

**Uncertainty proxy.** Let $h_i^{(N)}$ be post-$N$.sub activations for item $i$ in a batch of size $B$. We use

$$\widehat{\sigma}_N^2 \;=\; \frac{1}{B}\sum_{i=1}^{B}\big\|\mathrm{GAP}(h_i^{(N)}) - \tfrac{1}{B}\sum_{r=1}^{B}\mathrm{GAP}(h_r^{(N)})\big\|_2^2,$$

and maintain an EMA to obtain $\sigma_N^2$.

---

**Algorithm 2** Variance-triggered growth at the current leaf

---

**Require:** current leaf $L$, estimate $\sigma_L^2$, threshold $\theta_{\mathrm{grow}}$, child cap $M_{\mathrm{max}}$
 1: **if** $\sigma_L^2 > \theta_{\mathrm{grow}}$ **and** $|L.\mathrm{children}| < M_{\mathrm{max}}$ **then**
 2:     create one or more children $C$ using the same sub-module family as $L$
 3:     initialise child weights freshly; set $\alpha_C \leftarrow 0$
 4:     attach each $C$ to $L.\mathrm{children}$
 5: **end if**

---

## 4.3 Dynamic Pruning

To bound compute and depth, a global pruning sweep runs every $E_{\mathrm{prune}}$ epochs. Children with persistently low usage are disabled (or removed).

---

**Algorithm 3** EMA-based pruning sweep

---

**Require:** node $N$ (skip root), threshold $\theta_{\mathrm{prune}}$, minimum age $t_{\mathrm{min}}$
 1: **for all** child $C \in N.\mathrm{children}$ **do**
 2:     **if** $\alpha_C < \theta_{\mathrm{prune}}$ **for the last** $t_{\mathrm{min}}$ **epochs then**
 3:         mark $C$ inactive (and optionally delete from $N.\mathrm{children}$)
 4:     **end if**
 5:     PRUNESWEEP($C$)
 6: **end for**

---

# 5 Methodology

This section describes how we evaluate *Juliet*: (i) test its applicability and FLOP savings without accuracy loss, (ii) position it against dynamic SOTA (SkipNet (73), ConvNet-AIG (**?** ), BlockDrop (74)), and (iii) probe which components matter via ablations.

## 5.1 Versatility and Applicability

**Why CNNs?** We target convolutional networks to provide a mature, well-understood testbed with clear compute patterns and strong baselines. CNNs enable precise FLOP/accuracy measurement and fast iteration relative to transformers, while still exhibiting hierarchical features that align with Juliet's adaptive computation.

**Backbones.** We instrument three families to span a design spectrum: *ResNet-18* (residual blocks; easy to skip/prune; standard dynamic baseline); *EfficientNet-B0* (compound scaling + SE; stress-tests whether Juliet adds efficiency atop hand-optimized designs); and *DenseNet-121* (dense connectivity; stresses bandwidth/reuse; checks routing beyond simple block boundaries).

**Protocol (CIFAR-10).** For each backbone and its Juliet variant, we train for 15, 30, or 60 epochs using identical pipelines. Table 2 shows the hyperparameters used for the experiments. We log accuracy, theoretical FLOPs, memory usage, wall time, and the dominant ops. CIFAR-10 provides sufficient permutation space at modest cost. Training runs on a single CPU core on ARCHER2; GPUs are reserved for ImageNet in §5.2.

Table 2: CIFAR-10 hyperparameters

| Hyperparameter | DenseNet-Juliet-121 | EfficientNet-Juliet-B0 | ResNet-Juliet-18 |
|---|---|---|---|
| Batch / Split / Aug | 128; 90/10; crop 32+4, flip, norm | same | same |
| Max Trie Depth | 4 | 4 | 4 |
| Init Channels / Growth | 64 / 32 | 32 / 16, width= 1.0 | 64 / N/A |
| Epochs | 15,30,60 | 15,30,60 | 15,30,60 |
| Optim / LR / WD | Adam / 1e-3 / 1e-5 | same | same |
| Scheduler (mode/factor/patience) | ReduceLROnPlateau (min/0.5/3) | same | same |
| Grow / Prune | $\theta_{grow}$=0.4; every 3 ep; $\theta_{prune}$=0.1 | same | same |

**FLOPs & energy measurement** Inference FLOPs are computed with THOP on a dummy input (CIFAR-10: $1\times3\times32\times32$). Training FLOPs/epoch are approximated as $3\times$ inference FLOPs $\times|\mathcal{D}_{train}|$. Energy is not collected for CPU-only CIFAR-10 runs.

## 5.2 Positioning Juliet Among Dynamic Architectures

We compare Juliet to three prominent dynamic methods on ImageNet: *SkipNet*, which uses layerwise gates to decide whether to execute or bypass residual blocks per sample (73); *ConvNet-AIG*, which features soft Bernoulli gates trained via straight-through estimators for tunable depth (72); and *BlockDrop*, which employs an RL policy to learn block-keep/drop decisions under a FLOP budget (74).

All models use ResNet-101 backbones adapted per method; Juliet uses ResNet-Juliet-101. Training runs for 120 epochs on an NVIDIA H100 (80 GiB HBM3) at EIDF.

**Metrics.** Top-1/Top-5 accuracy, training/inference FLOPs, energy (NVML joules), throughput (imgs/s) per node, distribution of executed nodes, and per-class traversal depths. Figure **??** shows the setup for the ImageNet experiments.

Table 3: ImageNet training setup (ResNet-Juliet-101).

| | |
|---|---|
| Batch / Workers / Pin | 256 / 8 / True |
| Train aug | RandResizedCrop 224, flip, norm$(0.485, 0.456, 0.406)/(0.229, 0.224, 0.225)$ |
| Val prep | Resize short=256, center crop 224, same norm |
| Optim / LR / Mom / WD | SGD / 0.1 / 0.9 / 1e-4 |
| Scheduler | CosineAnnealingLR ($T_{max}$=40) |
| Trie params | Grow thr. 0.50; prune every 5 ep; prune thr. 0.10; grow every 10 mini-batches |
| Device | CUDA (TF32 enabled if available) |

**FLOPs & energy on ImageNet.** Inference FLOPs via THOP at $1\times3\times224\times224$. Energy integrates NVML power per mini-batch using trapezoidal rule; values reported per epoch for train/val. If NVML unavailable, entries are NaN with availability status.

### 5.3 Understanding Juliet: Ablations & Instrumentation

We isolate contributions on CIFAR-10 using ResNet-Juliet-101 (chosen for depth/modularity, and for well-documented FLOP/latency). Each ablation runs 40 epochs on one ARCHER2 CPU core.

**Ablations.** We isolate contributions through several ablations: we test the *selector architecture* by replacing the transformer with a 2-layer MLP or a depthwise-conv+GAP module; we compare routing policies (deterministic top-1 vs. stochastic sampling with temperature annealing); we analyze the growth/prune mechanics by testing pruning-only, growth-only, and disabled (static Juliet) modes to measure impacts on capacity allocation and generalization; and we conduct a 50-trial Optuna sweep for hyperparameters (depth, grow/prune thresholds/interval, LR, and WD).

**Software stack.** PyTorch 2.3.0 (57), Python 3.12.3 (70), `torchvision` for IO/aug (53). AMP (`-amp`), `torch.compile`, Optuna orchestrates sweeps externally (3).

**Profiling outputs.** We emit a static profile (`model_profile.json`: dataset, input size, params, FLOPs, proxy flag, NVML availability) and per-epoch logs (`epoch_{k}.json`: losses; top-1/5; train FLOPs/epoch; timings; energy; LR; per-node throughput; Juliet node execution counts). Optional analyses (node-usage histograms; Grad-CAM sparsity stats) run when available.

Initialisation across backbones)

Initialisation for **ResNet-18/Juliet** starts with 5 nodes (root+depths 1–4), growing and pruning from data. **DenseNet-121/Juliet** also starts with 5 nodes, aligned to its 4 dense blocks and transitions, growing dense sub-branches on high variance and pruning idle paths. **EfficientNet-B0/Juliet** uses a max depth of 4 (root+4) with selectors at stage boundaries. **ResNet-101/Juliet** prebuilds 10 nodes, with subsequent structure learned via growth and pruning.

**Computation of FLOPs/energy** Let $F_{\text{inf}}$ be per-example inference FLOPs (THOP). Training FLOPs per epoch $\approx 3 F_{\text{inf}} |\mathcal{D}_{\text{train}}|$. For energy on GPU, per-epoch joules $J \approx \sum_t \frac{\Delta t}{2}(P_{t-1} + P_t)$ with $P_t$ from NVML.

## 6 Results & Analysis

### 6.1 CIFAR-10 on ARCHER2 (CPU)

Juliet cuts inference FLOPs by $\sim$**21%** (ResNet-18), $\sim$**68%** (EfficientNet-B0), and $\sim$**70%** (DenseNet-121); training FLOPs fall correspondingly under matched schedules (e.g., DenseNet $\sim$70% at 60 epochs). Largest gains occur in over-parameterized backbones. See Table 4 for the derived data from the experiments.

Table 4: Parameter counts, compute, and accuracy for baseline vs. Juliet (trie) variants.

| Family | Variant | Params (M) | Inference FLOPs (M) | Training FLOPs (T) | Final Test Accuracy (%) |
|---|---|---|---|---|---|
| EfficientNet | EfficientNet-Juliet-B0 | 0.27 | 41.96 | 339.88 | 84.93 |
| | EfficientNet-B0 | 7.16 | 126.62 | 1074.26 | 91.83 |
| DenseNet | DenseNet-Juliet-121 | 3.20 | 17.79 | 144.07 | 85.27 |
| | DenseNet-121 | 6.96 | 59.11 | 478.81 | 85.88 |
| ResNet | ResNet-Juliet-18 | 19.68 | 29.36 | 237.78 | 83.73 |
| | ResNet-18 | 11.18 | 37.22 | 301.48 | 85.04 |

Accuracy is **within $\sim$1%** of baseline for ResNet-18 and DenseNet-121; EfficientNet-B0 loses a few points under aggressive pruning. Juliet is most effective where baseline slack exists (DenseNet); compact backbones (EfficientNet-B0) need gentler pruning.

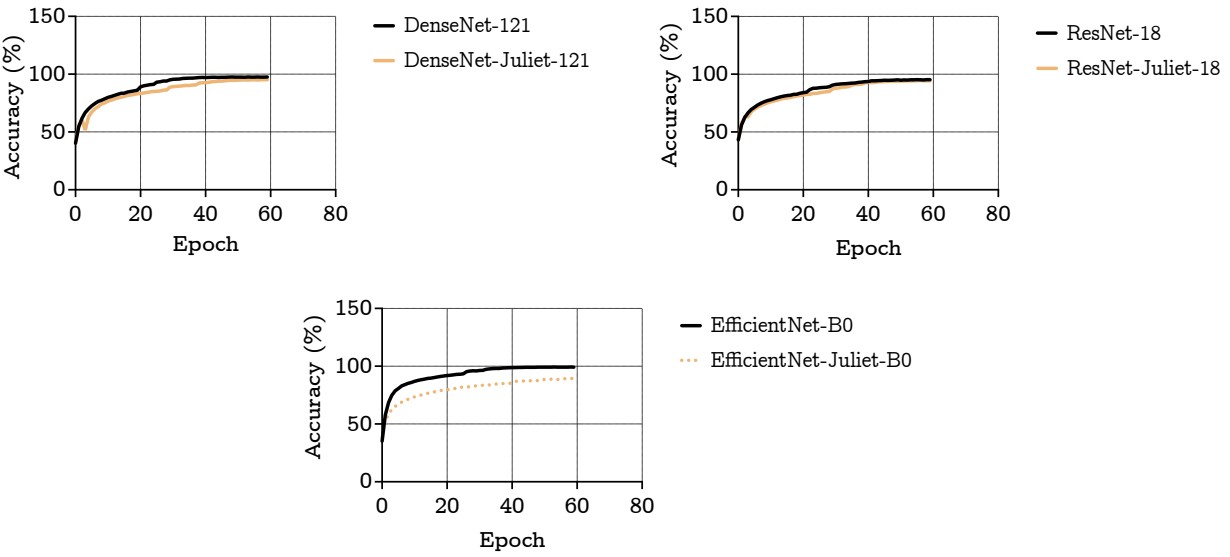

Figure 2: Training accuracy curves for ResNet-18, DenseNet-121 and EfficientNet-B0 over 60 epochs (includes baseline vs its Juliet variant

Prune/grow events delay early learning; Juliet often *catches up* by longer training. In our runs, baselines reached ∼85% by 30 epochs; Juliet variants typically matched by 60 epochs. Early stopping penalizes Juliet.

**Observation**

On CIFAR-10, Juliet achieves substantial FLOP cuts with minor accuracy loss; allow longer schedules to stabilise after pruning.

Lightweight *depthwise+GAP* and *2-layer MLP* routers matched transformer-router accuracy; the transformer is *helpful but not required* at this scale.

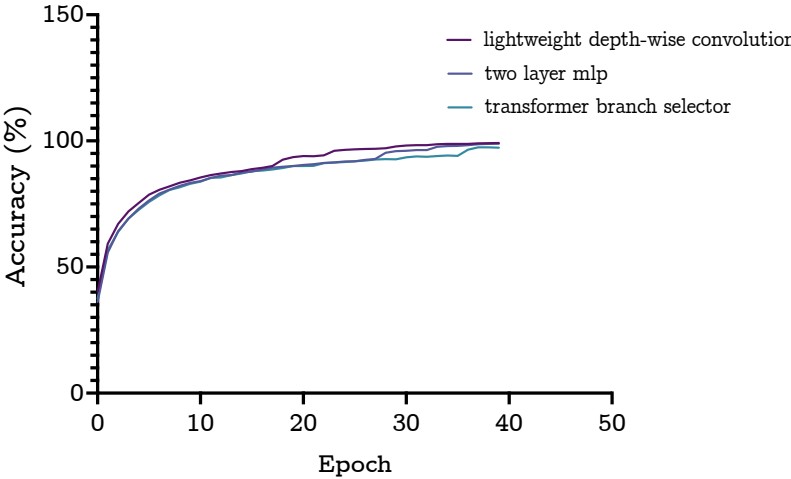

Figure 3: Training accuracy curves for different model architectures over 40 epochs. The plot compares lightweight depth-wise convolution, two-layer MLP, and Transformer branch selector models.

We profiled end-to-end time and memory (beyond theoretical FLOPs). Dynamic routing overhead is small, but bandwidth/SIMD limits cap speedups.

Table 5: CPU profiler snapshots (averaged across slices; PyTorch CPU profiler).

| Model | Metric | Static | Juliet | Notes |
|---|---|---|---|---|
| **ResNet-18** | Wall time | 0.882 s | 0.963 s | Juliet +9.2%; conv-bwd share $\sim 66\% \rightarrow 60\%$ |
| | Peak CPU memory | 188.55 MB | 265.33 MB | +40.7% (routing overhead) |
| **DenseNet-121** | Wall time | 2.573 s | 1.239 s | **−51.8%** (fewer backward convs) |
| | Peak CPU memory | 521.50 MB | 133.70 MB | **−74.4%** |
| **EfficientNet-B0** | Wall time | 4.424 s | 4.774 s | Juliet +7.9%; conv-bwd $\sim 46.1\% \rightarrow 41.2\%$ |
| | Peak CPU memory | 9.54 GiB | 9.96 GiB | +4.4% |

**Key Takeaway**

FLOP cuts do not map linearly to speedups on CPU: depthwise convs are bandwidth/SIMD limited; DenseNet benefits most (redundancy, memory relief); ResNet gains are modest; EfficientNet shows faster steps despite higher nominal FLOPs (skipped heavy bwd regions).

**Observation**

Theoretical FLOP savings sometimes understate/overstate wall-time gains; memory bandwidth and SIMD utilisation dominate CPU performance.

## 6.2 Hyperparameter Search (CIFAR-10, ARCHER2)

**Cost-aware objective.**

$$\mathbf{L} = (1 - \alpha) + \lambda c$$

where $\alpha$ is accuracy and $c$ is the normalised compute cost (FLOPs proxy). The hyperparameter $\lambda$ balances accuracy against compute efficiency.

**Optuna (50 trials).** We vary: maximum depth, grow threshold, pruning interval, pruning threshold, learning rate, and weight decay.

**Best Trial**

**depth = 2**, **growth threshold $\approx 0.91$**, **prune interval = 2**, **prune threshold $\approx 0.65$**.

These settings converged to a compact architecture achieving *mid-80s* validation accuracy and an *order-of-magnitude* FLOP reduction (e.g., $\sim$5e8 vs. $\sim$7.8e9 on a ResNet-101 scaffold). A high grow threshold combined with early regular pruning stabilised the topology rapidly.

**Observation**

Under a compute-aware loss, Juliet prefers small, efficient models; accuracy per FLOP improves markedly.

-

## 6.3 ImageNet on NVIDIA H100 (ResNet-101 variants)

We compare **Juliet-101** to dynamic ResNet-101 baselines: SkipNet (73), BlockDrop (74), ConvNet-AIG (**?** ).

**Setup.** 120 epochs on H100 (80 GiB). Juliet prebuilds 10 nodes; grows on feature-variance triggers; prunes idle branches periodically.

**Results.** **Juliet-101: 72.20%** top-1 at **2.66 GFLOPs/image**. Static ResNet-101 ∼76.4–77.4% @ ∼7.6 GFLOPs.

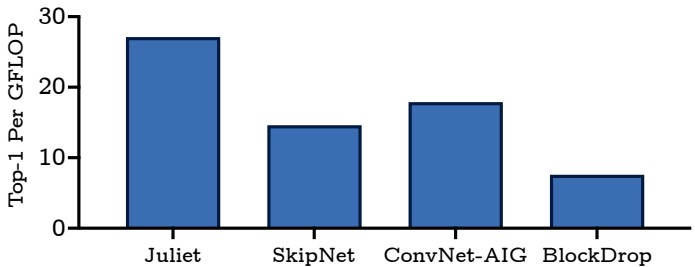

Figure 4: Top-1 accuracy per GFLOP, highlighting Juliet's computational efficiency.

**Comparative accuracy (top-1):** SkipNet-101 achieves **77.37%** (∼5.32 GFLOPs) (73); ConvNet-AIG-101 reaches **77.22–77.37%** (∼4.31–5.08 GFLOPs) (**?** ); and BlockDrop-101 scores **76.80%** (∼14.7 GFLOPs default) (74). Juliet lags by **4.6–5.3** points, but uses the *least* compute.

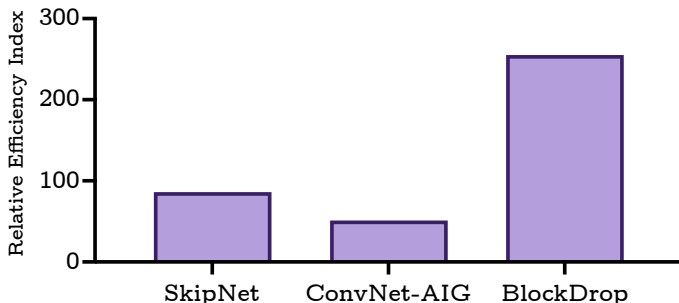

Figure 5: Relative efficiency index comparing Juliet with other SOTA dynamic networks.

**Efficiency.** Top-1 per GFLOPs: **Juliet 27.1**, SkipNet 14.6, ConvNet-AIG 17.9, BlockDrop 7.6. Juliet is **+86% / +51% / +255%** more efficient respectively.

Training energy ∼0.33–0.36 J/img (NVML); Top-1 rises steadily, energy per image remains flat; inference compute stays near-constant across epochs. Juliet's grow–prune cycles in deep nets can slow convergence.

To close the accuracy gap: *(i)* prebuild 15–25 nodes; *(ii)* more conservative growth and slightly more permissive pruning; *(iii)* lengthen prune intervals to reduce oscillations.

> **Key Takeaway**
>
> ResNet-Juliet-101 is *Pareto-efficient*: no compared method achieves a better Top 1 per GLOPs. With calmer growth/pruning and deeper initialisation, accuracy should rise without sacrificing efficiency.

### 6.4 Hardware–Software Insights

We observed several hardware-software insights. First, FLOPs ≠ time on CPU, as memory bandwidth limits and depthwise SIMD underutilisation cap speedups; DenseNet benefits most via memory relief. Second, regarding SIMD/parallelism, depthwise convs underuse AVX-512-like lanes; Juliet helps mainly by *skipping*

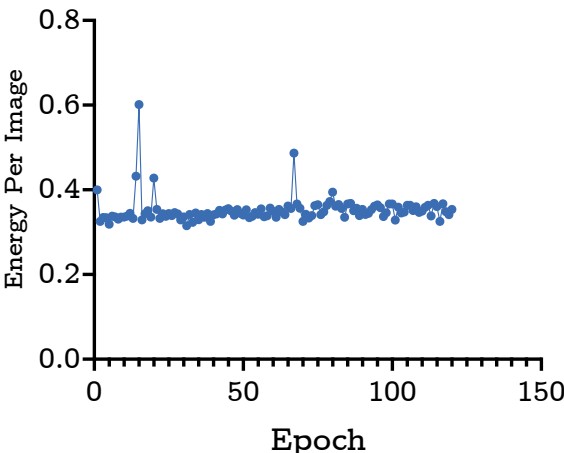

Figure 6: Energy consumption profile during model training on ImageNet.

such ops, not by improving their efficiency. Third, cache locality improves mildly as Juliet's trie narrows fan-out and reuse horizons, with the biggest effect in DenseNet. Fourth, batch divergence from input-dependent paths had a minor impact in our CPU profiling but can matter on accelerators. Finally, regarding param vs. working set, Juliet may add parameters for routing, but only the active path participates per input, keeping inference working sets bounded.

### 6.5 Synthesised Insights & Tuning Guidance

Juliet is most advantageous when *inference efficiency matters* and models have redundancy (DenseNet-like). For compact backbones (EfficientNet-B0), use milder pruning and longer training.

Practical knobs include tuning the *growth threshold*, *prune threshold*, *prune interval*, and *prebuild limit* to the deployment point. For edge/latency/energy constrained deployments, one should use higher grow/prune thresholds, longer prune intervals, and modest prebuild. For accuracy-first scenarios, lower thresholds and shorter intervals with a larger prebuild are appropriate, though one must monitor for instability. Run small Pareto sweeps (Top-1, GFLOPs, latency, memory) and pick the knee.

> **Key Takeaway**
>
> Juliet is a tuneable, compute-aware pathway to efficient inference: pay a little more training, save a lot at deployment—often without meaningful accuracy loss.

## 7 Conclusion

*Juliet* demonstrates strong versatility across convolutional neural networks, consistently reducing theoretical FLOPs while maintaining competitive accuracy and favorable memory behavior across multiple CNN backbones, achieving the best Top-1 per GFLOPs among the baselines we compared, surpassing methods such as SkipNet. While profiling confirms a gap between theoretical savings and wall-clock speed due to routing overhead, ablations (on CIFAR-10) reveal that *Juliet* naturally evolves compact topologies and that expensive transformer routers are not essential, as lightweight (MLP+conv) perform comparably. Key future directions focus on practical deployment and efficiency gains: developing lighter, more stable routing policies (e.g., RL-based gating) to reduce overhead; scaling via distributed computing; and pursuing hardware-software co-design using compilers (e.g., TVM, TensorRT) or specialized hardware (e.g., FPGAs/ASICs). Further research should also address integration with quantization and pruning, generalization to new modalities (e.g., NLP and speech), security vulnerabilities introduced by dynamic routing (e.g., timing side channels),

the development of formal theoretical bounds, and leveraging the model's inherent hierarchical structure for interpretability.

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
