# OpenReview forum: "Juliet: Per-Sample Conditional Branching for Efficient Con- volutional Networks"
_TMLR — Rejected by TMLR_

### Review · Reviewer_Lojc · 2026-01-02

**Summary Of Contributions:**

The paper introduces a hierarchical architecture where layers are organized as nodes in a trie that enables prefix sharing and flexible branching. It employs a routing mechanism at each node using lightweight, transformer-based selectors that direct data flows based on intermediate feature representations. Finally, the work proposes a grow-prune framework that evolves model capacity online through variance-driven expansion triggers and usage-based pruning of underutilized nodes.

**Audience:**

No

**Audience Explanation:**

While the problem addressed is itself very relevant within the TMLR community, the current state of the paper renders it insufficiently clear to be valuable to the community. While the finding might be interesting, it would require enormous rewriting and additional experiments to have a potential impact (see comments on the  claims and comments below)

**Broader Impact Concerns:**

No ethical concerns.

**Claims And Evidence:**

No

**Claims Explanation:**

- **Clarity and Presentation:** The paper lacks the pedagogical depth necessary to make the proposed method fully understandable or reproducible (see details provided below). The narrative is often unconvincing and lacks technical precision. Furthermore, the writing style appears to have been heavily post-processed by an LLM, resulting in a text that is difficult to follow and occasionally obscures the underlying technical details. In its current state, I am unable to provide a solid assessment of the methodology as the quality of the writing renders the technical details nearly impossible to fully understand. The paper requires significant revision to improve clarity and transparency before its scientific merit can be properly evaluated.
- **Outdated Baselines and Limited Empirical Scope:** While the paper addresses a highly relevant topic, the empirical evaluation is insufficient. The proposed method is compared exclusively against methods from 2018, which makes the claims regarding state-of-the-art performance unconvincing. Additionally, the experiments are largely confined to CIFAR-10; to demonstrate the method's value, evaluation on more complex datasets and diverse tasks beyond image classification (such as NLP or image segmentation) is required.
- **Architectural Constraints:** The evaluation is performed solely on CNN architectures. The justification for this focus is not compelling, and without experiments on modern architectures (e.g., Transformers), the potential impact and generalizability of the paper remain minor.
- **Superficial Ablation Studies:** The ablation analysis is quite limited and does not isolate the impact of the core components. For example, the specific contribution of the trie-structure is not evaluated. Additionally, the authors mention a "practical training recipe that stabilizes topology learning," but the paper fails to explain what specific mechanisms ensure this stability under standard supervised losses.
- **Methodological Transparency:** There is significant ambiguity regarding the experimental protocol. It is not clear how the hyperparameter search was conducted or whether a dedicated validation set was utilized to ensure results are not overfitted to the test set.

**Requested Changes:**

Below is a list of requested changes. In its current state, the low clarity of the paper makes it difficult to assess the technical novelty or the validity of the provided solution. Consequently, addressing these changes may not be sufficient on its own, as new technical questions are likely to arise once the manuscript becomes readable.

- Restructure the manuscript to create a cohesive narrative between Sections 2 and 5, ensuring that the notations and equations introduced in the background are actually utilized in the technical description and motivation.
- Revise Section 3 to be more than a list of existing works; each paragraph should explicitly relate the cited literature to the proposed method and explain how this work addresses specific limitations in the field.
- Expand the introduction to include a proper survey of the field and more than a single reference, as the current high-level overview fails to contextualize the paper's contributions.
- Clarify Figure 1 by defining the visual encoding, including the meaning of each color, the function of the yellow diamond, and whether the diagram illustrates a single block or the global structure.
- Provide a formal mathematical description of the trie structure, its specific implementation in this context, and the mechanics of the branch selector.
- Include a clear distinction between the procedures performed at training time versus inference time, ideally supported by a flow diagram or pseudocode.
- Replace imprecise, "script-style" phrasing and "big words" with accurate technical definitions; specifically, define terms such as "underpowered gates," "kernel launch costs," and "irregular control flow."
- Redesign the notation list on page 7 to be understandable, providing clear definitions and the motivation behind each symbol used.
- Reorganize the manuscript so that the "Methodology" section focuses on the technical contributions rather than the evaluation protocol, in line with standard machine learning literature.
- Eliminate pseudo-sentences and informal shorthand (e.g., "FLOPs != time on CPU") in favor of professional, complete technical arguments.
- Clarify the specifics of the "University of Edinburgh’s system" and provide a technical justification for why this specific environment was chosen for benchmarking.
- Include a detailed discussion of existing work on trie-augmented networks and explicitly state the technical novelty of the proposed method relative to those works.
- Remove or replace the profiling outputs and software stack descriptions with more informative content, such as a link to the source code and a reproducible environment configuration.
- Expand the "Notes" column in Table 5 to ensure the information is legible and not overly condensed.


Given the depth of the structural, empirical, and stylistic issues, I am skeptical that the manuscript can be revised sufficiently to meet the required standard for a second round of review.

---

### Review · Reviewer_ExUz · 2026-01-14

**Summary Of Contributions:**

This paper introduces a dynamic neural architecture named Juliet. Juliet uses a trie-based structure in which each node is a predefined block of layers. Each node is also equipped with a transformer-based router. At each node, the router selects the next child node (a block of layers) based on the current node's features. As Juliet is a trie-structured augmented approach, the obtained structure can grow or be pruned.  Whenever uncertainty in a leaf node passes a certain threshold, a new node is created. When the exponential moving average (EMA) of a node is less than the threshold, that node and its children are pruned.

Strength:
1. The novel approach for dynamic neural architecture is proposed by augmenting tree- based architecture.
2. The suggested pruning and growth methods are conceptually simple and do not require pretraining or offline architecture search
3.  Reduction of Flop on CIFAR-10 on CPU by 21%, 68% and 70% for ResNet-18, EfficientNet-B0, and DenseNet-121.

Weakness:
1. The writing of the paper does not follow the principles of academic writing. Many references are not attached to citations, such as in page 12 ", ConvNet-AIG (? )". Figures are not referenced in the main text, which makes connecting paragraphs to the corresponding figures troublesome (see page 13).
2. Paper structure creates confusion, for example, section 3 starts with existing work, then it jumps to the proposed method, Juliet, as a subsection of section 3.
3.  Many concepts are introduced without citing references, for example, the exponential moving average (EMA).

4. The notation is inconsistent, and variables in the equations are not properly introduced. For example, when EMA is introduced in the paper (Equation 3), beta is used as a hyperparameter. However, later in Algorithm 1, lambda is used as a hyperparameter for a similar purpose. Or in section 4.2, the function GAP is not introduced.

5.  The experiment protocol varies substantially across datasets and experiments. As a result, it is so difficult to follow the results, compare different outcomes, and interpret them.

6. There are several inconsistencies in the experiment section. For example, the paper switches backbones inconsistently: ResNet-18 is used as the primary backbone for CIFAR-10 experiments, but then switches to ResNet-101 for CIFAR-10 ablation studies without adequate justification.  Since ResNet-101 is overparameterized for CIFAR-10, the ablation result may not be representative of the behavior observed in the main experiments.  To ensure consistency of ablation studies, the same backbone as the primary experiment must be used.

**Audience:**

Yes

**Audience Explanation:**

This paper would be interesting for researchers working on dynamic neural networks and conditional computing with application. However, in the current format, with inconsistencies in the experimental protocol and a gap between claimed and demonstrated results, it does not provide reproducible data or knowledge for further studies.

**Broader Impact Concerns:**

There is no broader impact concern.

**Claims And Evidence:**

No

**Claims Explanation:**

Unfortunately, the claims are not supported by experiments.

1. Insufficient justification for hardware bottlenecks: In section 6.4, the authors mentioned "FLOPs ̸ = time on CPU, as memory bandwidth limits and depth-wise SIMD under utilization cap speedups". However, no empirical justification, such as measuring cache hit/miss statistics, SIMD utilization rates, or memory bandwidth, is provided to support this claim.
2. Practical inefficiency on standard architecture: for 2 out of 3 evaluated backbones (ResNet-18 and EfficientNet-B0), the Juliet version of these architectures resulted in higher wall-clock time. While the framework works well on highly redundant architectures such as DenseNet-121, these results suggest that their method's gain comes primarily from removing redundancy rather than effective conditional computing.
3. Experiments do not support the use of a transformer-based router.  In Figure 3, they showed that a 2-layer MLP performs almost identically to the proposed transformer selector, which contradicts the need for such a complex router.
4. The paper claims that it provides a dynamic architecture for deep models. However, in their hyperparameter tuning results in section 6.2, depth 2 is optimal. So, this makes Juliet's approach collapse into a mixture-of-experts (MOE) model, which raises the question of whether this complex branching model is useful.

**Requested Changes:**

1. Clearly separate Juliet architecture from existing work in Section 3 to clearly distinguish the proposed method from prior approaches.
2. Ensure all citations are correctly linked to references.
3. Use the consistent notation and terminology throughout the paper. In particular, use consistent symbols for the same hyperparameters throughout a paper, or, if a change is required, establish the relation between the new and old symbols.
4.  Use the consistent experiment protocol across experiments. If different backbones or datasets are used, the reason must be justified. For example, ablation studies should not use substantially more complex backbones (e.g., ResNet) on simpler datasets (e.g., CIFAR-10) without proper justification. Either conduct ablation experiments on ImageNet or maintain a model with a proper level of complexity for CIFAR-10.

5. Provide empirical evidence for the benefit of introducing a transformer-based router: Currently, the results do not show any advantage in using a transformer-based router. Demonstrating the same results on a more complex dataset, such as ImageNet, can be more beneficial.

6. Align the claim of the paper to experimental evidence: Ensure to refer to figures in the corresponding text.
7. Demonstrate the necessity of deeper architecture and the benefits of trie-based architecture.  At least one experiment (Hyperparameter experiment). must show that the proposed approach requires a deeper architecture and does not simply collapse to MOE.

---

### Review · Reviewer_udMv · 2026-01-16

**Summary Of Contributions:**

The method introduced, termed “Juliet”, is a dynamical neural network approach that implements both training- and inference-time algorithms for conditional sample-wise routing through submodules. The goal is to reduce the FLOP count of models so they run more efficiently on computationally limited devices. The system is structured as a trie, where samples can traverse the various submodules along the trie based on necessity, as determined by a router at each trie node. Per node, the router outputs probabilities for each child node, and a sample is sent deterministically to the child node with the highest probability. Each trie node contains both a submodule, e.g., a ResNet block, and a router, such as an attention-based, MLP, or convolutional approach.

Algorithmically, this has two implementations, dependent on whether it is training or inference time.

At inference time, a batch is passed to a root node, which outputs its usual activations for the block, which a router then assigns a child node per sample for further computation. Each sample is assigned to its top (argmax) probability determined by the router. Hence, this is a deterministic approach, with the implication of reducing anomalous circumstances in which a sample is probabilistically sent to a less-favourable node (with a stated ablation performed against a stochastic choice). For each sample in the child node, this process repeats, descending through the graph until reaching a leaf node, which is then classified.

Training time employs the same routing approach, but with periodic trie pruning and growth. One or more child nodes are added to the sample's current leaf node if the activation variance of that leaf exceeds a threshold; the variance is computed across channel-wise activations following from a global average pooling.  Provided the maximum number of child nodes has not been reached, a child node is then added that is the same module as the parent, but freshly initialised. Pruning occurs when the router has consistently not routed samples through the child node for a substantial time – this is assessed using a thresholded EMA, accumulated across every run, of how many times it is routed to that node. This is not performed solely at leaf nodes, but at every node in the system except the root.

Several different architectures are considered and tested for the router: an attention-based encoder module, a convolutional module or an MLP. Provided to these are a ‘weight signature’ for each child node and a feature vector of channels from GAP applied to the node’s submodule output, which is then postprocessed with LayerNorm and possibly a channelwise adapter if needed. This is then used to route the sample further. The empirical results indicate that the MLP performs optimally, although the manuscript appears to suggest that attention is the preferable default.

**Additional Comments:**

The manuscript includes identifying information, namely “University of Edinburgh”, which should have been redacted to “University of [Anonymised]” to prevent reviewers from inferring the author or research group, or from conveying any institutional favourability/association, thereby enabling biased reviews. Moreover, information on computer ownership is not important to reviewers or scientifically necessary, especially when computational specifications are already provided in the text for “H100”, etc. This extends to “EIDF” & “ARCHER,” which should have been redacted as well. These may not constitute a hard violation of anonymity, but they are details that are inappropriate/unnecessary to include, and they are at the edge of the intended ethos of a double-blind review.

**Audience:**

Yes

**Audience Explanation:**

Overall, *this is a valuable and novel methodology to add to the literature*, especially with the proposed trie neural architecture and node-based routers. This trie-based organisation appears well-motivated, largely novel (although reference 1 first introduces it) at the submodule level, and hence practically significant. The approach is also well motivated and contextualised in the early section, comparing differing dynamic network approaches.

However, the results are currently insufficient to position the work clearly against existing methodologies. Several requested changes are critical for acceptance, necessitating more overall clarity for readers rather than being SOTA contingent. The novel approach would certainly remain valuable even if the mean and standard deviation indicate lower optimality than current top-1 metrics suggest, and the addition of these is strongly encouraged. More in-depth discussions are also needed to help practitioners understand any training-time/inference-time failure modes. Such considerations need to be explicit and clear for prospective readers interested in using this approach.

As the paper currently stands, I **wouldn't be comfortable drawing the community's attention to this work due to the identified flaws**. However, with all the proposed corrections, I believe this methodology would be a valuable addition to the TMLR community and of interest.

**Claims And Evidence:**

No

**Claims Explanation:**

Empirical results are gathered across a variety of modern convolutional architectures, namely ResNet, DenseNets, and EfficientNet, comparing accuracy, theoretical and practical FLOP counts, CPU time, memory, and several other metrics. The flop reductions are supported by evidence, and the method achieves significant CPU savings for DenseNet, although the model incurs higher CPU time for ResNet and EfficientNet. Inference and training FLOPs are decreased across these models, with notable parameter reductions for EfficientNet and DenseNet, but significantly higher for ResNet. All of these models are well justified by indicating how each tests specific aspects of the methodology. Similarly, GPU tests are run on ResNet-101 for Juliet ablations against SkipNet, ConvNet-AIG, and BlockDrop, limited to top-1 accuracy and FLOPs.

However, the manuscript **currently quotes only top-1 accuracy across models**, which is indicative of performance, but statistical measures are severely lacking throughout the work. The mean and standard deviation of performance would provide readers with a clearer impression of the reliability and expected performance of this approach, positioned against existing approaches in the literature. Although Top-1 performance is valuable for production models, it should be used alongside standard statistical metrics for a more holistic view. Similarly, all other metrics are quoted and plotted without error estimates across independent repeats (including independently retrained networks). Given the computational efficiency of the method demonstrated throughout the manuscript, such repeats to provide thorough statistics would not appear troublesome to compute.

The **absence of these metrics obfuscates potential training-time/inference-time failure modes** and their likelihood, which are inevitably present to some degree even if rare. However, any failure modes are not clear from the Top-1 accuracy and are not explicitly discussed. This is not in reference to Tables 4 and 5, where Juliet performs suboptimally for ResNet/EfficientNet compared to other models, which is explicitly discussed, but instead the failure modes which arise when using the Juliet methodology in isolation during training/inference - i.e. stalled training, width/depth limits, considerations, etc.

Furthermore, the manuscript **rhetorically elevates attention as the preferable routing mechanism**, stating that “Transformers yield higher-fidelity decisions,” “transformer is helpful but not required at this scale”, and “Juliet’s transformer-based selector (§3.1.2) can attend to richer context, improving decision quality” **yet this claim is unsupported by empirical evidence**, which, contradictorily, indicates that it is the worst-performing of those tested, albeit close in performance to MLPs and Convolution forms. Hence, these statements suggesting its optimality are in conflict with the given results.

The first and third quotes cite “Attention is all you need,” with the placement of the first quote’s citation suggesting that the paper supports the stated claim. Yet "attention is all you need" doesn't evaluate attention-based conditional routing architectures; instead, it presents the general transformer architecture, which uses attention for context-based token refinement. Hence, “Transformers yield higher-fidelity decisions” and “improving decision quality” **are neither supported by the manuscript's empirical results nor attributable to the cited paper**. Overall, current empirical results suggest that Juliet does not benefit from a transformer-based router, even if conceptually appealing, and the paper should note this is speculative/conceptual rather than a supported result. All the places comments explicitly endorse that transformers require correction, and a concluding statement should instead grapple with the conceptual and speculative preferability of attention.

Finally, architectures are known to benefit substantially from batched inputs with GPU acceleration; however, most results report CPU-based performance, which will disproportionately favour the Juliet model over the baselines. This is because Juliet uses sample-wise routing and hence typically forfeits computation across batches, but the cost of this is obscured by CPU results, as the baseline does not exhibit its typically batched advantage. This, in itself, is fine if the manuscript emphasised the use of Juliet in cases where only a CPU is available, but this context is absent when these specific results are discussed. It is acknowledged that “batch divergence [...] had a minor impact in our CPU profiling but can matter on accelerators”, but this is a single sentence on page 14. **This needs to be foregrounded for reader clarity**. It could be easily remedied by contextualising these considerations alongside those results in the same discussion and by stating them upfront in the abstract as a clear motivation when hardware is limited.

It is acknowledged that an H100 was used to collect several ablation results, but crucially, these results provide only accuracy and FLOP rate, which are invariant to these batch accelerators since they are model-routing specific. Hence, **GPU runs should have at least measured runtime similarly to CPU metrics**, and secondly, accuracy per time is arguably a more appropriate metric (particularly since it is noted that FLOPs do not indicate runtime and batched acceleration isn’t reflected in FLOPs but in time). These should also be statistically analysed across the datasets/runs, with mean and stddev for GPU FLOPs & time as well.

**Requested Changes:**

These are ordered from most critical to least critical; however, all are important considerations for acceptance.

1) **Critical**: As discussed, current empirical results report top-1 metrics and generally metrics with no distributional information. This distributional information should be present for in-text, tabulated, and graphed results. To make it clear, this applies to all quantities stated: RAM usage, CPU time, and every empirical numeric provided. This is a critical oversight and does not give readers a fair impression of average performance, variability, or the method's brittleness. It also does not indicate any failure modes of the architecture besides comparative analyses. The requested change is to include primarily mean +/- stddev results primarily, and the existing top-1 results secondarily. Any poorer-performing results should then be discussed, and their failure modes interpreted to give readers greater context and insight into the practicalities of the approach. The statistical metrics are especially important in this case, since the approach is strongly sample-specific, and top FLOP reductions and CPU time could arise in very shallow instances evaluated in the architecture. It needs to be clear that these statistical measurements are taken across the testing dataset. Providing these statistical results will therefore offer a more transparent understanding of the approach's successes and pitfalls, as well as where the methodology is most applicable. These should all be explicitly discussed in detail. (To add: even having underperformance in these added metrics will not detract from an early-stage and novel methodology, as future development in the research direction is expected and welcomed for such work; however, without these, the work remains poorly positioned and not transparent in its comparison to predecessors.)

2) **Critical**: The manuscript repeatedly states that attention routers are preferable, with other results being comparable but not optimal; however, the results presented contradict this claim. The alternative routers outperform attention in the plot. All statements similar to those previously mentioned should be corrected to align with the empirical results, and correct citations/citation amendments should be made for clarity. If the author(s) wish to conceptually support the attention implementation, this should be highlighted as speculative, contradictory to current results, and perhaps relegated to conclusions/future work discussions, as the results primarily do not assert that attention is preferable. The plots showing router performance (Figure 3) also require mean ± stddev.

3) **Critical**: Provide an upfront, comprehensive discussion of sample-wise routing and how it forfeits batched advantages. The addition of GPU runtime is necessary to fairly reflect the impact of accelerators on batched results and to provide context on how Juliet performs relative to standard practice. This can be used to contextualise when the Juliet model should be used in practice. The manuscript presents this only in CPU tests, a regime that gives Juliet an unfair and unrealistic advantage because of the CPU-based loss of batched accelerators. If this CPU-based regime is the central motivating hardware constraint, then it is fair, but it should be clearly foregrounded, e.g., in the abstract, introduction, and conclusions. GPU time rather than FLOPs will provide a fairer benchmark and evaluation regime for each model's performance. Overall, these considerations should be upfront and ideally clarified in GPU benchmarks, or Juliet should be clearly presented as a primarily CPU-based approach.

4) **Major**: The results clearly indicate the RAM memory requirements for the models, but it is unclear whether this directly corresponds to the disk space required. If a trie architecture is used, where nodes tend to have numerous children, each with a similarly sized submodule, then presumably this will require substantially higher disk footprint than the original sequential model being adapted. This is significant information for readers. It is unclear whether these nodes are held on disk until being loaded into RAM for routing evaluation. Such a procedure would result in a large discrepancy between hardrive and the stated memory metrics - and it is presently unclear how the two relate. Therefore, this needs to be explicitly discussed, and if the disk footprint and RAM metrics differ, then both need to be stated concurrently and clearly. This is especially true if the approach is intended to run on computationally limited devices, where both hard drive and RAM requirements may be significant limitations.

5) **Major**: The accuracy per GFLOP, although insightful, is a more uncommon metric and is underdiscussed in the work. It disproportionately benefits Juliet as a metric in results and should be in addition to standard accuracy plots. It is suggested that Figure 4 displays both standard accuracy and accuracy per GFLOP concurrently to offer readers greater clarity into the approach. Accuracy per CPU/GPU-time would be important additional plots.

6) **Major**: Several design decisions are underdiscussed and are not justified within the text. They appear mostly as heuristic choices rather than well-motivated. For example, using the channel-wise activation variance metric is stated and not motivated. Alternatives do not appear to be discussed, e.g., quantifying information gain. Similarly, the channel adapter is underspecified, and its particular implementation is not motivated beyond its necessary presence. The compact weight representations are a crucial aspect of the routing model, but they are underspecified; alternatives are not considered, and a discussion of the advantages and disadvantages of each alternative signature is absent. Similarly, new child nodes replicating the parent with fresh initialisation are not justified in the text, nor are the specifics of initialisation provided. These are just some examples of underspecified/unmotivated design choices in the text, which require clearer justification and comparison with alternative implementations, considering their benefits and drawbacks.

7) **Major**: Greater exposition on which features of models benefit the most from Juliet. It appears particularly advantageous for DenseNet, but the manuscript would benefit greatly from a dedicated section that further discusses, in a clear and concise manner, the key architectural regimes where Juliet is most applicable and why. A dedicated section or within the conclusion would help clarify this. Similarly, a more detailed discussion of which architectural aspects make the approach less compatible and should be distilled from the empirical results, rather than just conceptual. Generalisable high-level takeaways would be useful upfront in the introduction, discussions and conclusions, so readers know when this approach is applicable.

8) **Major**: Provide a more in-depth discussion of why Juliet has a higher peak CPU-memory requirement, and other cases where it underperforms.

9) **Major**: Several further ablations are mentioned, but results and conclusions are absent. For example, but not limited to, stochastic routing versus deterministic. All such ablations should have results provided and a surrounding discussion, even if provided in the appendices.

10) **Major**: The proxy of training-time FLOPs is estimated at 3x inference and is currently unjustified. It should be thoroughly explained or computed appropriately.

11) **Minor**: Are there any code repositories and seeds available for fair replication of these results? Although specific identifiable GitHub links should be redacted for review, leaving in-text anonymised mentions of these would be a good addition and help readers evaluate the work independently.

12) **Minor**: Figure 3 scales to 150% accuracy and ends at 40 epochs, but the graph continues to 50 epochs. The graph axes should be rescaled to 100% accuracy and 40 epochs. Similar for Figure 2, which could use some work in general (also inconsistent line plotting for Juliet runs, off-centred central plots, etc.). Stylistically, these need refinement for reader clarity.

13) **Minor**: Both on pages 8, 9, 12 and 13, there are missing citations shown as a question mark. Similarly, “Figure ??” on page 9. Please fix all instances of these. If the project is in LaTeX, using Overleaf will display all such missing reference occurrences – all require correction.

14) **Minor**: I would really like to see at least a brief discussion of the implications of gradients arising from dynamic routing across different submodules. This will have an impact on training dynamics to some degree, and some discussion will help clarify them. For the purpose of this work, it can remain a high-level discussion, but it is felt that the manuscript would benefit from this. Some discussion of dynamic networks and gradients is present in section 2.2, but is not applied to the Juliet approach.

15) **Minor**: Figure 1 requires several edits; the text and boxes cross through the dashed line, making the diagram stylistically unappealing and difficult to read. Several arrows have unusual offsets (such as those from the feature variance monitor), and the heads of arrows appear without tails on Node 2A. Overall, this diagram needs substantial refinement for acceptance; it is largely unclear, and the caption is insufficiently descriptive. It also infers that the same transformer branch selector is used across all cases, but the text suggests that each node has its own router - this needs clarification.

16) **Minor**: The pain points on page 6 are stated as names, but each is minimally discussed. A clear statement of the identifying characteristics, a brief discussion of each, and an explanation of when and where they occur and how the proposed method mitigates them should be provided.

17) **Minor**: In Table 1, the column “Trie structure” is a redundant discriminant between models, as it is largely a novel implementation in this manuscript; hence, of course, predecessors didn’t feature it (the table should perhaps include the architecture from the first reference, which introduces trie networks). Also, comment c of this table is unclear to readers, as it can be confused with comment a due to the line break.

18) **Minor**: Provide an explanation of the naming convention “Juliet”. It does not appear to be an acronym of any form. It would be beneficial to understand the etymology and significance of this choice.

19) **Minor**: Generally, captions are insufficient and could use more detail (including a brief description of visuals to improve accessibility for readers, such as for TTS systems).

20) **Minor**: Some refinement to the abstract phrasing “while growing and pruning capacity on the fly”, to indicate this is a training time adaptation. The abstract is used for readers to assess the method's relevance to their task, and greater precision in this phrasing will help them decide its relevance to their problem upfront.

---

### Decision · Action_Editor_6bU7 · 2026-02-28

**Recommendation:** Reject

**Audience:**

No

**Audience Explanation:**

While initially 2 out of 3 of the reviews answered "yes" to this question, at the official recommendation stage 2 out of 3 reviewers answered "no" and again I note the lack of response (no rebuttal), in view of which it is not possible to make a case for this paper.

**Claims And Evidence:**

No

**Claims Explanation:**

Reviewers unanimously indicated that this submission failed to support claims with accurate, convincing and clear evidence. I note also the lack of rebuttals, indicating implicitly the author's acceptance of the reviews. The reviewers have shared constructive feedback which can be of great help in re-shaping and strengthening the manuscript for a future submission.